# Barriers to and Facilitators of the Implementation of a Micronutrient Powder Program for Children: A Systematic Review Based on the Consolidated Framework for Implementation Research

**DOI:** 10.3390/nu15245073

**Published:** 2023-12-12

**Authors:** Yinuo Sun, Jiyan Ma, Xiaolin Wei, Jingya Dong, Shishi Wu, Yangmu Huang

**Affiliations:** 1School of Public Health, Peking University, Xueyuan Rd, No. 38, Beijing 100181, China; ynsun@stu.pku.edu.cn (Y.S.); jiyma@bjmu.edu.cn (J.M.); 1610120113@pku.edu.cn (J.D.); 2Dalla Lana School of Public Health, University of Toronto, Toronto, ON M5T 3M7, Canada; xiaolin.wei@utoronto.ca (X.W.); shishi.wu@utoronto.ca (S.W.)

**Keywords:** micronutrient powder, implementation science, barriers and facilitators

## Abstract

Background: As one of the most cost-effective investments for improving child nutrition, micronutrient powder (MNP) has been widely used in many countries to underpin the Sustainable Development Goals, yet challenges remain regarding its implementation on a large scale. However, few studies have explored the factors that facilitate or impede the implementation process using implementation science theories and frameworks. To address this gap, we adopted the Consolidated Framework of Implementation Research (CFIR) and conducted a systematic review of studies on the implementation barriers to and facilitators of MNP interventions. Method: Five publication databases, including EMBASE, Medline, PubMed, Web of Science, and Scopus, were searched for studies on the influencing factors of MNP interventions. Based on the CFIR framework, the facilitators and barriers for the MNP program implementation reported in the included studies were extracted and synthesized by five domains: intervention characteristics, outer setting, inner setting, individual characteristics, and process. Results: A total of 50 articles were eligible for synthesis. The majority of the studies were conducted in lower-middle-income countries (52%) through the free delivery model (78%). The inner setting construct was the most prominently reported factor influencing implementation, specifically including available resources (e.g., irregular or insufficient MNP supply), structural characteristics (e.g., public-driven community-based approach), and access to information and knowledge (e.g., lack of training for primary-level workers). The facilitators of the engagement of private sectors, external guidelines, and regular program monitoring were also highlighted. On the contrary, monotonous tastes and occasional side effects impede intervention implementation. Additionally, we found that the inner setting had an interrelation with other contributing factors in the MNP program implementation. Conclusion: Our results suggest that MNP program implementation was prominently influenced by the available resources, organizational structure, and knowledge of both providers and users. Mobilizing local MNP suppliers, engaging public-driven free models in conjunction with market-based channels, and strengthening the training for primary-level health workers could facilitate MNP interventions.

## 1. Background

Child nutrition is a critical part of global health and human development, contributing to the achievement of several Sustainable Development Goals (SDGs), specifically SDG 2 (to end hunger and all forms of malnutrition) and SDG 3 (to ensure healthy lives and promote well-being for all at all ages). Malnutrition in early childhood can impair the development of children’s immune systems, increase morbidity and mortality, and cause severe cognitive and psychomotor development delays with long-term consequences [1]. Despite advancements in social and economic development, the prevalence of child undernutrition remains high, particularly in low- and middle-income countries (LMICs), where the existing health system infrastructure is weak and access to life-saving interventions has been interrupted by the COVID-19 pandemic [2,3,4]. By 2020, the global stunting, underweight, and wasting prevalence rates were 22.0%, 5.7%, and 6.7%, respectively [5]. However, one-third of all undernourished children globally reside in sub-Saharan Africa, and the prevalence rates of stunting, underweight, and wasting were, respectively, 38.5%, 25%, and 9% [6].

Micronutrient supplementation or fortification in the “1000 days” (or from conception to two years old) is widely recognized as a window of opportunity for preventing malnutrition and its consequences. For children aged 6–23 months, vitamins and minerals need to be fortified in the diet to ensure physical and mental development and prevent morbidity and mortality. Many countries have implemented nutrition intervention programs to improve children’s overall nutritional statuses, such as by distributing powder, syrups/drops, foodlets, pills, or capsules containing iron, zinc, or micronutrients. However, interventions involving syrups/drops, pills, or capsules often face challenges in terms of poor adherence due to administration difficulties [7]. As a low-cost substitute product with the benefits of easy administration, comprehensive effects, and safety, the home-fortified version of micronutrient powder (MNP) has been widely recommended for large-scale nutrient interventions by the World Health Organization (WHO) [8]. According to the first global guideline for evidence-based MNP formulation introduced by WHO, the MNP is a tasteless powder containing iron, vitamin A, and zinc, either with or without other micronutrients, targeted at children aged 6–23 months and children aged 2–12 years [9]. Substantial evidence supports the efficacy of MNP interventions, as the consumption of a daily MNP supplement effectively reduces the prevalence of wasting, underweight, stunting, and anemia [10].

An increasing number of countries have adopted the WHO’s guidelines and scaled up MNP programs for children aged 6–23 months or 36 months. Between 2011 and 2020, the number of countries implementing MNP programs increased from 22 to 57 countries, most of which are LMICs [11]. Current models for distributing MNP include (1) free or subsidized distribution by the public sector as part of ongoing health programs, (2) subsidized distribution by the private sector as part of social marketing programs, or (3) free distribution by non-governmental organizations in emergency contexts [12]. Several challenges have been posed on the scale-up of an MNP program, such as the lack of access to MNP, the low acceptance of MNP among children, and the inappropriate usage of MNP [12]. A more comprehensive understanding of these factors is essential to maximize the health benefits and promote the scale-up of MNP interventions. However, existing relevant systematic reviews mainly focused on evaluating the outcomes and effectiveness of MNP programs, with few reviews systematically examining factors that facilitate or impede the implementation process from the perspective of implementation science [13,14]. 

To address this knowledge gap, we adopted an implementation research approach to identify the barriers and facilitators for MNP implementation systematically. We subsequently provided recommendations for improving MNP coverage and adherence in order to address the micronutrient deficiencies of children.

## 2. Methods

### 2.1. Conceptual Framework

This review used the CFIR framework to analyze the implementation barriers and facilitators documented in the reviewed studies. The CFIR is a determinant meta-theoretical framework that positively or negatively influences program implementation based on published theories and models [15,16]. It specifies 39 constructs across five domains: the characteristics of the organization implementing the intervention, factors external to the organization, characteristics of the intervention, characteristics of the individuals involved in the implementation, and the process of implementation [17]. 

### 2.2. Search Strategy

The search was conducted in five databases up to 13 February 2023, including EMBASE, Medline, PubMed, Web of Science, and Scopus. The keywords of the research scope were based on concepts and domains, including target populations, intervention types, and settings. Specific search strategies used for each database can be found in Appendix A. We searched all relevant peer-reviewed journal articles and excluded gray literature.

### 2.3. Study Screening and Eligibility

The inclusion and exclusion criteria are shown in Table 1. We conducted a two-step process to identify the ones that meet the inclusion criteria. The first step was to create a list of studies without duplicates through the five databases. The second step included a two-round screening. During the first round, titles and abstracts were screened by two researchers separately. If the information was unavailable for decision, the study was moved to the subsequent full-text review as a part of the second round of eligibility screening. During full-text screening, we excluded the studies that were editorials, commentary pieces, and systematic reviews; interventions that were not targeted at children aged 6–23 months or 36 months; and articles that were not in English or Chinese. Articles not obtained through online databases and library searches were excluded from the final analysis. After two rounds of screening, 50 articles were moved forward for data extraction.

Two researchers performed a quality appraisal of the included studies independently. Studies selected for data extraction were then assessed using the Mixed Methods Appraisal Tool (MMAT). This critical appraisal tool appraises the methodological quality of five categories of studies: qualitative research, randomized controlled trials, non-randomized studies, quantitative descriptive studies, and mixed-methods studies [18]. Each category of studies was assessed against five criteria, and reviewers rated each criterion with “Yes”, “No”, or “Can’t tell”. 

### 2.4. Data Extraction

Two researchers extracted data from the eligible papers. Data extraction forms included study title, authors, publication year, country of study, study aim, study design or method, target population, and description of the intervention reported in the study. Implementation barriers and facilitators in the original text were directly extracted from each study and coded according to the construct’s definition of the CFIR framework. Any disagreement or modifications to construct definitions were discussed among the two reviewers until a consensus was reached. Additionally, since the CFIR does not specify the interactions between those constructs, we identify and explain the relationships between the constructs within and across the domains. The review was completed following the Preferred Reporting for Systematic Reviews and Meta-Analyses (PRISMA) statement and checklist (Appendix A). 

### 2.5. Search Results and Included Studies

A total of 9111 citations were returned from databases, and 3470 duplicates were removed (Figure 1). After we screened titles and abstracts, 5538 publications were excluded. We conducted full-text screening for 93 articles and included 50 in the final review. The PRISMA flowchart is shown in Figure 1. 

## 3. Results

### 3.1. Study Characteristics

A total of 50 studies published from 2003 to 2021 were included in the final analysis, and the articles’ characteristics are summarized in Table 2. We sorted the studies by country according to the World Bank’s 2023 country classification. Most studies were conducted in LMICs (n = 26, 52%). In 78% (n = 39) of the 50 studies, MNP interventions were delivered in a free distribution manner by the public sector, while only a small portion (n = 11, 22%) were market-based implementations. Regarding the study population, 27 involved only users, and 24 included both providers and users. The study designs were classified into randomized control trials (28%, n = 14), mixed-methods studies (31%, n = 16), cross-sectional studies (20%, n = 10), qualitative studies (18%, n = 9), and a case study (2%, n = 1). The detailed information of each study is summarized in Appendix A.

The non-response bias was also high across most studies that applied descriptive cross-sectional methods, while most qualitative studies had relatively fewer biases. In many mixed-methods studies, the divergences and inconsistencies between the quantitative and qualitative results were not adequately addressed. One case study was not assessed using the mixed-methods appraisal tool. Appendix A provides a description of the quality assessment of each study.

### 3.2. Barriers and Facilitators 

The extracted barriers and facilitators in five domains and 23 constructs are shown in Table 3. We only present the common factors that occurred in at least five studies to ensure consistency.

### 3.3. Domain 1: Intervention Characteristics

The first major domain of the CFIR is related to the characteristics of the intervention being implemented. In this domain, we identified evidence strength, design quality, packaging, and cost as influencing factors.

Eleven studies cited evidence strength and quality as factors related to MNP implementation [3,12,19,20,21,22,23,24]. The commonly reported facilitator was the adoption of formative research and pilot studies, which provide evidence for program development. For example, countries like China have introduced critical public health interventions nationally after a series of pilot programs. Madagascar [25] and Kenya [21] have completed preparatory phases of formative and pilot research and are now scaling up MNP distribution. 

Five studies have reported that MNP possessed several advantages over other nutritional supplements. As for the health benefits, MNP provides micronutrients simultaneously, which is more effective than single-nutrient sprinkles [22,26]. In terms of cost, because of their composition, weight, and size, MNP products are less expensive to produce, transport, and store compared to centrally processed and fortified complementary foods [26,27]. In addition, mixing MNP into food to deliver vitamins and minerals is easier compared with using iron drops or syrup [19,26,28].

Twenty-one studies reported the design quality and packaging as barriers to the compliance of MNP, including a monotonous taste and a change in the smell of the food [11,29,30,31,32,33,34,35,36,37,38,39], a lack of information on the ingredient base of the packaging [21,29], and occasional negative side effects [19,29,31,36,37,38,40,41,42]. Notwithstanding the convenience to use or store the product, the attractive appearance and the improved taste were observed to improve the acceptability of MNP [24,31,34,43,44,45]. 

Nine studies addressed cost as a barrier to implementing MNP interventions. Product cost was the most common challenge cited in the market-based delivery setting, where the users failed to afford the products [30,32,33,40,45]. Also, indirect costs, including transportation distance and transportation time, were reported in three studies [43,46,47]. Furthermore, one study illustrated that the improved appetites stressed household finances because of increased food costs [45]. 

### 3.4. Domain 2: Outer Setting

The outer setting includes the economic, political, and social contexts within which an organization resides [22]. Outer organizational structures and external policies were observed to be particularly influential in MNP program implementation.

Cosmopolitanism, specifically the partnerships and coordination among key stakeholders in the private and public sectors, was articulated to influence MNP implementation in eight studies [3,20,22,29,31,42,46,48]. These studies highlighted the private sector’s role in program implementation. Specifically, it provided MNP products and matched services, including quality control and social marketing.

Thirteen studies addressed facilitators to external policies and incentives [3,12,19,22,23,27,29,33,40,43,46,47,49,50,51]. The WHO and other international organizations have introduced global guidelines. These guidelines have explicitly recommended the MNP ingredients, dose, and proper administration and outlined the implementation steps of the MNP program. Accordingly, local stakeholders have enacted enabling guidelines, policies, and regulations to support MNP design and program implementation. However, a gap exists between the guidelines for use and successful operation and execution. Interventional guidelines are not country-specific, so they need to be adapted to the local context first to help implementation in countries that are newly conducting MNP interventions [12,22]. 

### 3.5. Domain 3: Inner Setting

The inner setting encompasses features of structural, political, and cultural contexts through which the implementation process will proceed. Three constructs, including structural characteristics, available resources, and access to information, were widely cited as factors in the inner setting domain.

The available resource was reported as a barrier in 12 studies. Irregular or insufficient MNP supply was a common challenge facing program sustainability. In several countries, unavailable MNP was generated by a lack of local suppliers and unstable supply chains. Several studies also cited that adequate allocation of financial and human resources was essential but limited in many countries [12,23,29,30,40,45,46,50,52]. For example, primary-level workers were often too busy to perform intervention activities. However, there were no financial incentives for the added tasks in most countries [12,19,27,29,42].

Structure characteristics were reported as facilitators in 10 studies. The most common driving factor was the good coordination of the MNP program within the health system. Relying on the primary health care system, primary-level workers undertook the tasks of MNP distribution, social and behavior change communication (SBCC), and program monitoring (such as making regular visits to evaluate if MNPs were used correctly and effectively) [3,20,31,39,44,45,46,52,53,54]. In this way, the users were more receptive to interventions considering the authority of primary-level workers. In contrast, structure constraints included the extra burden for the health systems, mainly extra work for health care providers on top of their heavy daily work [19,28,46]. 

Fourteen studies identified the access to knowledge as an influencing factor. The researchers acknowledged that regular training for primary-level workers promotes MNP interventions, for they delivered counseling knowledge to the caregivers while distributing YYB. [19,31,46,53]. However, the training for primary-level workers was insufficient and inaccurate in numerous countries [3,21,23,32,40,51,52,55]. As a result, the validity of the information they provided would be decreased due to the absence of systematic training for primary-level workers [3,21,23,29,34].

### 3.6. Domain 4: Characteristics of the Individuals Involved

How individuals involved in MNP interventions perceived and acknowledged the program highly affected its implementation [23]. The commonly detected characteristics of individuals include personal knowledge and beliefs, the individual stage of change, and other personal attributes. Twenty-four studies acknowledged barriers in knowledge and beliefs about the intervention of caregivers, among which ten studies positively reported facilitators in this construct. Specifically, caregivers’ perception of children’s nutritional improvement highly improved the intervention adherence [11,24,28,32,36,41,44,46,54,56]. However, the lack of awareness and inadequate knowledge of the product in caregivers prevailed [21,30,31,37,43,45,46]. Additionally, caregivers were not adequately aware of the proper use of MNP, which, in part, led to the poor taste of the MNP [22,57]. In addition, some target caregivers were worried about the occasionally occurring side effects of MNP [3,12,29,34,43,50,52]. 

Six studies focus on defining the stage at which an individual progresses toward proficient, enthusiastic, and long-term intervention usage. The factors in the construct of the individual stage of change were recognized as facilitators. In four studies, caregivers were reluctant to accept MNP as they did not gain peer support from neighbors or family members [29,30,32,34]. However, even after receiving support from family members or neighbors, the caregivers were unlikely to change their attitudes towards MNP on account of insufficient awareness [25,50].

In fourteen studies, other personal attributes, including the low educational levels of caregivers, the lack of time or ignorance of MNP use, and the inappropriate initiation of complementary feeding, were addressed as barriers [32,34,35,36,38,40,41,52,54,58]. One case suggested that a mother’s older age was found to motivate the successful implementation of MNP interventions [43].

### 3.7. Domain 5: The Process of Implementation

Successful implementation usually requires productive approaches and processes. We describe the two essential activities of engaging and evaluating as facilitators during the implementation.

Social and behavior change communication (SBCC) and social marketing are two main processes used to inform and engage individuals. SBCC for users on how to use MNP and resolve the side effects of MNP were productive to its implementation [3,19,23,27,28,33,34,35,36,37,47,50,54,56]. Social marketing and advocacy by private sectors in a market-based setting also play crucial roles in encouraging consumers to use MNP [3,20,31,46,48]. Nevertheless, three studies cited insufficient social marketing and mobilization as barriers to participant engagement, mainly when the MNP was sold to targeted beneficiaries [21,23].

The monitoring of the distribution, usage, adherence, and effects of MNP greatly affected its implementation as a facilitator in eight studies [3,19,23,31,40,48,54,59]. Six studies reported that routine home visits by primary-level workers (PWs) to monitor its implementation were conducive to the appropriate use of the MNPs. Other studies highlighted the effectiveness of regular monitoring by the evaluation team to trace the improvement in the children’s nutrition. However, the insufficient capacity for monitoring MNP interventions by health workers has led to challenges in maintaining regular monitoring [27,33,60,61].

### 3.8. Relationship between Constructs

The line between each setting is not always clear. Usually, changes in the other settings influence implementation mediating through changes in the inner setting [15]. We explored the interrelationship between several constructs that impact MNP program implementation, as shown in Figure 2. For instance, the training for health providers (inner setting) has facilitated the process of SBCC among users (process), which improved caregivers’ knowledge and belief towards MNP (individual characteristics) [19,20,56]. Additionally, community health workers (inner setting) played core roles in the processes of delivering the information, reflecting, and monitoring (process) [32,61]. The global and local explicit guidelines (outer setting) guaranteed the MNP design quality (intervention characteristics) [33,51]. 

## 4. Discussion

Our study systematically synthesizes barriers and facilitators of implementing MNP interventions across the five domains of the CFIR framework. The results indicated that the inner setting constructs, such as the available resources, structural characteristics, and access to information and knowledge, are common factors influencing the implementation of MNP programs. Additionally, the MNP taste and side effects (intervention characteristics) and caregivers’ beliefs about the interventions (individual characteristics) were commonly observed barriers. Coordination between private and public sectors (outer setting domain), external guidelines (outer setting domain) and program monitoring (process) were reported as facilitators in the studies. Moreover, in the market-based delivery model, the product cost (intervention characteristics) and insufficient market engagement (process) strongly negatively influenced the implementation. Finally, the inner setting constructs were observed to be highly interrelated with common factors in other domains. 

The lack of a sustainable supply of MNP was a commonly reported barrier to MNP interventions. Children’s nutrition was highly neglected in some underdeveloped countries despite the promising benefits of nutrition interventions for children in a window of opportunity [62]. As a result of the chronically underfunded investment of domestic budgets and international donor resourcing, MNPs have yet to achieve the desired coverage [63]. Therefore, securing the supply of MNP is key to the successful implementation and scale-up of MNP programs. Globally, the current MNP supply landscape is dominated by a few global manufacturers, which account for a 90% share of the sales volumes. However, few local producers exist in the current market, partly due to poor technology or a lack of incentives [63]. For example, only one manufacturer provides MNPs in Bangladesh. Hence, ensuring continuous supplies in many countries depends on procurement and import, which take complicated operations and interminable time. In contrast, some countries have facilitated a self-sufficient supply of MNP, such as China [64], where the local manufacturers embraced high-quality and low-cost MNP for the interventions. The WHO has included MNPs in the 2019 update of the Essential Medicine List (EML) [65], which suggests that MNPs should be considered essential products for public health impact. It is also a step towards making MNPs more accessible to needy children and ensuring greater cost-effectiveness in delivery through health systems. In this context, local suppliers play crucial roles in securing MNP supply and enhancing the MNP program by positively influencing consumers’ acceptance.

The community-based delivery approach in both public and private-driven models plays a vital role in the implementation of MNP programs. Community-based interventions generally refer to health interventions delivered by front-line health workers aiming to promote the well-being of people in a defined local community [66]. Our result highlights the positive effect of the community model in implementing large-scale health programs in low-resource settings. To extend the reach of interventions to all target groups, a strategy of decentralization can be employed, whereby the responsibility of delivery and health education is delegated to primary-level workers assigned to specific families. Moreover, the authority of the community primary-level workers was enhanced in this model, which helps improve the acceptance of intervention among the targeted population. In contrast, several challenges remained in the community-based models. For example, in highly federalized countries such as India and Nigeria, MNP interventions would require particularly intensive management and engagement with state-level policymakers and implementing bodies [67]. It was also suggested that community-driven MNP programs are often integrated into existing health activities, which can impose an additional workload on already strained health systems [46]. For primary-level workers, monetary and nonmonetary incentives can have critical motivating effects on their performance if deployed strategically. However, as demonstrated by our results, concerns about work overload and inadequate financial compensation have impeded the implementation of MNP programs, emphasizing the need to strengthen supporting incentives.

Although MNP was proven to be safe and effective, the effectiveness of home fortification with MNP depends on how it is used and accepted. Informative SBCC for caregivers was prominently productive in enhancing the adherence and acceptance of MNP, which echoed Eileen’s findings [68]. The behavior change messages play significant roles as follows: first, if misused, the tasteless powder may change the texture, color, and taste of the food slightly, discouraging children from eating foods with MNP; second, while the MNP remains safe for consumption and retains its nutritional value, the suspicion about its side effects, including occasional diarrhea or darkened stool, can negatively affect acceptability if caregivers are not informed that their child may experience side effects; finally, mixing a powder into a child’s food is a new behavior, and following a daily regimen is challenging [63]. In most countries, the delivery of MNP was coupled with the SBCC, including providing counseling and communication materials, like calendars and stickers, to caregivers [46]. Hence, there is a great need for health providers to provide information on side effects and proper administration for caregivers [9]. Training for health workers is, therefore, essential [69]. However, this research observed that a lack of refresher training for primary-level workers might lead to inconsistent, even inappropriate counseling for users [70], which suggests the need to further focus on training health workers who are involved in MNP programs.

In addition, we found that most countries distributed MNP free of charge via the health sector. In contrast, only a few countries adopted the distribution through a market-based approach, where primary-level workers sell the products to caregivers, such as Bangladesh Maternal and the Infant and Young Child Nutrition (MIYCN) home fortification program [71,72]. In addition to the above common factors, regardless of the delivery model, several challenges specifically existed in the fee-based model. MNPs tend to be sold at a price that is unaffordable for low-income families, and social marketing was insufficient. Currently, a mixed model where a free product and a fee-based product coexist can be a viable choice, as highlighted in Madagascar and in the study by Kenya et al. [25,40]. This approach effectively utilizes the coordination between the public and private sectors, thereby facilitating the expansion of the program’s reach.

### 4.1. Implications and Recommendations for Policy and Practice

Applying the CFIR framework, we captured the commonly reported implementation barriers and facilitators across LMICs and identified the interrelationship between the prominent themes. We concluded that a sustainable supply of MNP, the mobilization of community-based primary-level workers, and access to knowledge and information on MNP for providers and users play critical roles in the successful implementation of MNP programs. Therefore, we propose the following recommendations for policy formulation.

First, it is important to advocate for the global and local supply of MNPs to ensure the sustainability and potential scale-up of MNP programs. The local community should encourage the establishment of local MNP suppliers and support existing ones in contexts where they can impart benefits. In particular, large-scale manufacturers in countries with a high production capacity can provide technical and material assistance to newly developing manufacturers. The global community should promote access to MNP as a potential public good worldwide.

Second, adopting community-based channels to deliver MNP and perfect the incentives for primary-level workers is recommended. Primary-level workers should be employed, as they are the ones who deliver/sell, inform, motivate, and monitor usage among caregivers. Since the added work burdens have significantly impacted program implementation, it is consequently critical to establish an incentive mechanism for grassroots health workers to perform the task to improve their motivation, thereby ensuring the sustainable development of the program.

Third, strengthening ongoing training and supportive supervision for primary-level workers is needed to help disseminate knowledge on MNP and nutrition at the community level. The current SSBC in many countries has helped enhance the acceptance of MNP. To ensure that grassroots health workers provide accurate guidance, it is recommended to provide them with refresher training on counseling, accompanied by post-training assessments.

Finally, we recommend using public-driven free models in conjunction with commercial channels to scale-up MNP programs in lower-income countries. Given that improving nutrition for at-risk populations is a priority in many low-income countries, the government and the public sector should maintain a commitment to providing continuous nutrition interventions. Additional socially oriented and commercial channels can be essential in extending coverage and improving targeting on a larger scale.

### 4.2. Strengths and Limitations

This is the first review using the CFIR framework to systematically examine the influencing factors for implementing MNP interventions. Based on the CFIR constructs, we can comprehensively and systematically synthesize the barriers and facilitators from an implementation research perspective. In addition, we elaborated on the interrelationship between the identified factors. Ultimately, we provided recommendations for policy and practice to improve the sustainability of MNP programs.

There are some limitations to our study. First, we included studies in English and Chinese. The included studies covered the research on high-income, low-income, lower-middle-income, and upper-middle-income countries, which could ensure the representation of the result. However, we may have still missed some articles that were not published in English or Chinese. Second, although the included studies underwent quality appraisal, lower-quality studies were retained to allow for broad data capture, as we did not consider them to affect the representation of the results. Third, we included studies that were initially intended to explore the effectiveness of MNP, and not to document the barriers and facilitators.

## 5. Conclusions

Tackling the current challenges entails a focus on the available product, organizational structure, and access to knowledge, which were the main influencing factors for MNP program implementation. Promoting the production and supply of MNPs locally, engaging the public-driven community-based approach in coordination with market-based channels, and promoting training for providers who deliver the knowledge to users will maximize the reach of MNPs and improve the acceptability of MNPs.

## Figures and Tables

**Figure 1 nutrients-15-05073-f001:**
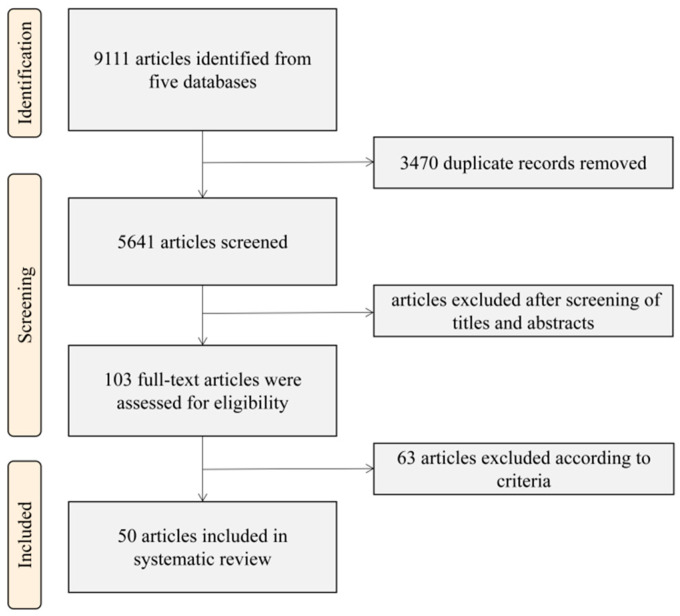
The PRISMA flowchart of the search and inclusion of studies.

**Figure 2 nutrients-15-05073-f002:**
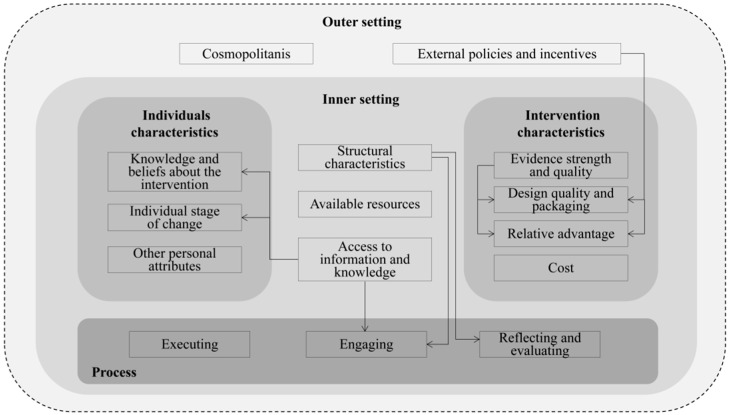
The interrelationship between CFIR constructs.

**Table 1 nutrients-15-05073-t001:** Inclusion and exclusion criteria.

Selection Criteria	Inclusion	Exclusion
Language	English or Chinese	All languages except English or Chinese
Study design	Original studies: qualitative, quantitative, randomized controlled trials, non-randomized trials, and mixed methods studies	Editorials, commentary pieces, and systematic reviews
Study population	Children aged 6–23 months or 6–36 months	Children in other age group
Study	Studies that reported implementation facilitators and barriers from data collection after implementing MNP interventions	Studies in which the facilitators and barriers were not reported

**Table 2 nutrients-15-05073-t002:** Included studies’ characteristics.

	Number of Studies	Percentage %
Summary by study region		
High income	1	2.0
Low income	9	18.0
Lower-middle income	26	52.0
Upper-middle income	9	18.0
Low and middle income	5	10.0
Summary by study population		
Providers	/	
Users	27	54.0
Providers and Users	23	46.0
Summary by study design		
Randomized controlled trial	14	28.0
Cross-sectional study	10	20.0
Mixed-methods study	16	32.0
Qualitative study	9	18.0
Case study	1	2.0
Summary by implementation settings		
Free distribution	39	78.0
Market-based	11	22.0

**Table 3 nutrients-15-05073-t003:** The specific barriers and facilitators that were reported in CFIR constructs that were addressed in studies as barriers or facilitators.

CFIR Framework Constructs	Barriers		Facilitators	
B. Cosmopolitanism			8	Partnerships and coordination between both private and public sectors
D. External policies and incentives	2	A gap exists between the guidelines for use and successful operational protocols	15	Global evidence-based guidelines
3. Inner setting				
A. Structural characteristics	3	Added burden on struggling health systems	10	Community-driven, decentralized, and integrated delivery approach
B. Networks and communications	2	Lack of coordination with communities and interagency coordination		
C. Culture	2	Insufficient attention to cultural situations, perceptions, routines, and practices		
D4. Organizational incentives and rewards	2	Health workers had low motivation to accept MNP distribution tasks, expressing concerns about work overload and inadequate financial compensation	1	
E2. Available resources	14	Interrupted or insufficient supply of MNP Insufficient staff Inadequate funding for MNP interventions	3	Sustainable availability and establishing various contact points for supplying MNP
E3. Access to information and knowledge	10	Absence of refresher training for frontline workers, inconsistent training on MNP counseling techniques, and lack of counseling materials or information	18	Positive experiences with training and supervision of primary-level workers
4. Characteristics of the individuals involved				
A. Knowledge and beliefs about the intervention	14	Lack of awareness and inadequate knowledge of MNP among caregivers Perceptions of side effects of MNP	10	Perceived positive changes in children following MNP use
B. Self-efficacy			1	Health workers’ confidence in ability to explain MNP benefits
C. Individual stage of change	4	Lack of familial and peer support for MNP use	2	Approval from family members Positive testimonies about the effectiveness of the MNP from relatives and neighbors
D. Individual identification with organization	1		3	Trust in the government and field staff
E. Other personal attributes	10	Low educational level of caregivers, lack of time or knowledge of MNP use in caregivers	1	Mother’s age > 25 years
5. The process of implementation				
A. Engaging	2	Inadequate communication regarding the health benefits and use of micronutrient powder for caregivers Insufficient social marketing and dissemination	6	Social and behavior change communication on how to use MNP and resolve side effects of MNP, home visits by community health workers Social encouragement and advocacy by private sectors
B1. Opinion leaders			1	Authority of health center staff
C. Executing	5	Complicated importation procedures and import taxes		
D. Reflecting and evaluating	5	Insufficient capacity for monitoring the program	8	Monitoring MNP distribution, usage, and adherence; evaluate the effectiveness of MNP program

## Data Availability

The following datasets were derived from multiple sources in the public domain: EMBASE, Medline, PubMed, Web of Science, and Scopus. All data that were generated or analyzed during this study are included in this published article and no new data were created.

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
