# Peer review of "Barriers to and Facilitators of the Implementation of a Micronutrient Powder Program for Children: A Systematic Review Based on the Consolidated Framework for Implementation Research"

_nutrients, 2023, doi:10.3390/nu15245073_

Round 1

Reviewer 1 Report

Comments and Suggestions for Authors

Children's Hunger and malnutrition are among the biggest problems in the modern world. The consequences of malnutrition are severe during the intensive development of the body. International organizations, including WHO, are trying to help prevent the tragic effects of malnutrition in the first years of life. There is a global consensus to focus on improving the coverage and quality of evidence-based interventions delivered at and around the time of birth.  One of the most effective ways is to provide the poorest countries with supplements containing vitamins and microelements - micronutrient powder MNP. WHO recommended using multiple micronutrient powders for point-of-use fortification of foods consumed by infants and young children aged 6–23 months and children aged 2–12. The success of any plan depends on the degree of its implementation.

 The authors of the work submitted for review attempted to evaluate the effectiveness of MNP supplementation programs. They conducted a systemic review of published studies on implementation barriers and facilitators using the adopted Consolidated Framework of Implementation Research (CFIR). The CFIR is an established implementation science meta-framework that initially included 39 relatively well-defined constructs organized around five conceptual domains: (1) intervention characteristics, (2) outer setting, (3) inner setting, (4) characteristics of individuals, and (5) implementation process. The search was conducted in five databases up to Feb 13, 2023, including EMBASE, Medline, PubMed, Web of Science, and Scopus. The final analysis included 50 studies published from 2003 to 2021:  randomized control trials, cross-sectional, mixed-methods, qualitative, and case studies. The inclusion and exclusion criteria were correctly described. 

The authors identified facilitators and barriers depending on the country, target groups, health system, and global or local manufacturers, community-based delivery compared to the market. They underlined the role of primary-level workers. 

The study was conducted very well following the applicable schemes. The information obtained was presented clearly, consistent with five main areas of analysis. Based on the information received, the authors developed recommendations for the increased effectiveness of further MNP  provision programs.

The work submitted for review was adequately planned, correctly performed, contains precisely formulated conclusions and, most importantly, provides information that will be useful in planning further programs regarding subsequent programs aimed at improving child nutrition.

 One uncertainty requires clarification

1. The study population included children aged 6-23 months or 6-36 months. Why not simply children aged 6-36 months? What was the reason for separating the two age groups? 

Author Response

Dear reviewer,

We appreciate your agreement on the value of this article, and your kindly advises in order to make this paper publishable. Thanks so much for your valued suggestions. We have answered all the comments point by point, and have revised our manuscript accordingly.

Reviewers' comments:

The work submitted for review was adequately planned, correctly performed, contains precisely formulated conclusions and, most importantly, provides information that will be useful in planning further programs regarding subsequent programs aimed at improving child nutrition.

One uncertainty requires clarification

  1. The study population included children aged 6-23 months or 6-36 months. Why not simply children aged 6-36 months? What was the reason forseparating the two age groups?
  • Response: Thanks a lot for your comment. The reason for separating the two age groupswas following: WHO recommends the use of MNP to improve iron statusand reduce anaemia among children aged 6–23 months, but the interventions in some countries have expanded age of targeted children to 36 months, such as the included study in Kyrgyz Republic, so we separate the two age groups for clarity.

Reviewer 2 Report

Comments and Suggestions for Authors

It is an interesting article, with a current and relevant topic since, as the authors point out, although it is known that child nutrition improves through the intake of micronutrients in powdered form, there is little analysis of the causes that facilitate or impede its implementation.

The following changes or improvements are advised:

Introduction:

Good introduction, well argued and with the necessary basic concepts to be developed regarding the research.

In line 62 and line 71, it is recommended that the date be updated to a more current one.

Methodology:

Very well argued, with sufficient explanation of the methodology used and the steps followed in the research. Maybe a little extensive, could be slightly reduced.

Results:

The first heading Search results and included studies (line 143 to 149) is advised to move to methodology.

Very good writing on Barriers and facilitators divided into five domains and 23 constructs in Table 3.

A good analysis of the results obtained.

Discussion:

Good argumentation on the subject matter and developed in depth.

Regarding the limitations of the study, it would be important to include studies in other languages, or at least to explain why not to include them, as has been done very well with the reasons for including studies of lower quality.

Conclusion:

Very interesting assessment of providers as enablers of success in these programmes and good indication for further studies.

It is recommended to update the bibliography, as the great part of it is not from recent years and, although in some cases it would not be necessary, for example in the citations related to WHO there are updates from August 2023 (https://www.who.int/tools/elena/interventions/micronutrientpowder-children).

Author Response

Dear reviewers,

We appreciate your agreement on the value of this article, and your kindly advises in order to make this paper publishable. Thanks so much for your valued suggestions. We have answered all the comments point by point, and have revised our manuscript accordingly.

Reviewers' comments:

It is an interesting article, with a current and relevant topic since, as the authors point out, although it is known that child nutrition improves through the intake of micronutrients in powdered form, there is little analysis of the causes that facilitate or impede its implementation.

The following changes or improvements are advised:

Introduction:

Good introduction, well argued and with the necessary basic concepts to be developed regarding the research.

In line 62 and line 71, it is recommended that the date be updated to a more current one.

  • Response: Thank you very much for your suggestions. We have updated the data and reference to a more current one.

Methodology:

Very well argued, with sufficient explanation of the methodology used and the steps followed in the research. Maybe a little extensive, could be slightly reduced.

  • Response: Thanks for your comment. We have simplified some explanation to make it more explicit.

Results:

The first heading Search results and included studies (line 143 to 149) is advised to move to methodology.

Very good writing on Barriers and facilitators divided into five domains and 23 constructs in Table 3.

A good analysis of the results obtained.

  • Response: Thanks a lot for your comment. We havemoved the first heading Search results and included studies (line 143 to 149)to methodology.

Discussion:

Good argumentation on the subject matter and developed in depth.

Regarding the limitations of the study, it would be important to include studies in other languages, or at least to explain why not to include them, as has been done very well with the reasons for including studies of lower quality.

  • Response: Thank you very much for your suggestions. We have revised and added the explanation as “we have included the studies in English and Chinese. The included studies covered the research on high income, low income, lower middle income and upper middle income countries, which could insure the representation of the result. However, we may still miss some articles published not in English or Chinese.”

Conclusion:

Very interesting assessment of providers as enablers of success in these programmes and good indication for further studies.

  • Response: Thanks a lot for your comment.

It is recommended to update the bibliography, as the great part of it is not from recent years and, although in some cases it would not be necessary, for example in the citations related to WHO there are updates from August 2023 (https://www.who.int/tools/elena/interventions/micronutrientpowder-children).

  • Response: Thank you very much for your suggestions. We have updated the bibliographywhich could be more current, and replaced the citations [9] with your recommended one.